# *Xenorhabdus* and *Photorhabdus* Metabolites for Fungal Biocontrol and Application in Soybean Seed Protection

**DOI:** 10.3390/jof11100691

**Published:** 2025-09-23

**Authors:** Nathalie Otoya-Martinez, Mustapha Touray, Harun Cimen, Edna Bode, Helge B. Bode, Selcuk Hazir, Julie Giovanna Chacon-Orozco, César Júnior Bueno, Luís Gárrigos Leite

**Affiliations:** 1Divisão Avançada de Pesquisa e Desenvolvimento em Sanidade Agropecuária do Instituto Biológico, Campinas 13101-680, SP, Brazil; notoya2501@gmail.com (N.O.-M.); cesar.bueno@sp.gov.br (C.J.B.); garrigos.leite@gmail.com (L.G.L.); 2Department of Biosciences, College of Science, Swansea University, Swansea SA2 8PP, UK; mtpha.touray@gmail.com; 3Recombinant DNA and Recombinant Protein Center, Aydin Adnan Menderes University, Aydin 09100, Türkiye; hcimen@adu.edu.tr; 4Department of Natural Products in Organismic Interactions, Max Planck Institute for Terrestrial Microbiology, 35043 Marburg, Germany; edna.bode@mpi-marburg.mpg.de (E.B.); helge.bode@mpi-marburg.mpg.de (H.B.B.); 5Molecular Biotechnology, Department of Biosciences, Goethe University Frankfurt, 60438 Frankfurt am Main, Germany; 6Center for Synthetic Microbiology (SYNMIKRO), Department of Chemistry, Phillips University Marburg, 35043 Marburg, Germany; 7Department of Biology, Faculty of Science, Aydin Adnan Menderes University, Aydin 09100, Türkiye; shazir@adu.edu.tr

**Keywords:** *Glycine max*, white mold, entomopathogenic nematodes, biological control, antifungal compounds

## Abstract

*Photorhabdus* and *Xenorhabdus* bacteria, members of the Morganellaceae family, are sources of novel natural products for the biocontrol of fungal pathogens in soybean production. This study demonstrated the inhibitory effects of metabolites from four *Photorhabdus* and *Xenorhabdus* strains (including a local isolate, *X. szentirmaii* PAM 25), against four key phytopathogenic fungi. Bacterial metabolite efficacy and fungal susceptibility varied. *Xenorhabdus szentirmaii* DSM 16338, *X. szentirmaii* PAM 25, and *X. doucetiae* demonstrated significant inhibition (>90%) against *Sclerotinia sclerotiorum*, *Botrytis cinerea*, and *Macrophomina phaseolina*, exhibiting superior efficacy compared to *X. nematophila* and *Photorhabdus kayaii*. *Fusarium oxysporum* demonstrated greater resistance to the bacterial supernatants. We identified fabclavine, pyrollizixenamide, and szentirazine from *X. szentirmaii*, and xenocoumacins from *X. doucetiae* as the antifungal bioactive compounds in the respective easyPACid mutants. Furthermore, we assessed the efficacy of *X. szentirmaii* PAM 25 and its metabolites in protecting soybean seeds from *S. sclerotiorum* and investigated the shelf stability of the bacterial metabolites as the fungus suppressors. Cell-free supernatant maintained >80% inhibition of *S. sclerotiorum* after one year at 5–35 °C. Importantly, the cell-free supernatant, as well as the bacterial culture, effectively inhibited *S. sclerotiorum* in seed treatments, ensuring ≥80% seed germination, comparable to thiophanate-methyl + fluazinam fungicide. This study demonstrates that the direct seed application of *Xenorhabdus* and *Photorhabdus* bacteria offers a practical and innovative biological control method against soil-borne fungal pathogens.

## 1. Introduction

Biotic stressors, particularly phytopathogenic fungi and oomycetes, continue to pose a significant threat to agriculture and forestry. These pathogens induce persistent crop and tree epidemics, resulting in substantial yield losses, estimated between 10 and 30%, and billions of dollars in annual economic losses [1,2]. Recent years have witnessed an alarming increase in the severity of disease outbreaks attributed to virulent oomycete and fungal pathogens. These organisms negatively impact global food security [3,4].

Notably, soybeans (*Glycine max*, Fam: Leguminosae), a crucial industrial crop valued for its high protein and oil content, are susceptible to several soil-borne fungi, including *Fusarium*, *Macrophomina*, and *Sclerotinia*. These pathogens, prevalent in leading soybean-producing regions, cause significant damage through seedling blight, seed decay, root and stem rot, charcoal rot, and blight or wilt [5,6,7]. For instance, Brazil has been the world’s leading soybean producer for the past five years, generating over USD 40 billion in yearly foreign sales; however, its crops remain under a continuous threat from persistent fungal and other diseases [8]. Furthermore, these fungi exhibit the capacity to persist in soil, infect plant tissues, and colonize crop residues, with their prevalence influenced by plant physiological state and environmental conditions [1,8,9,10].

Management strategies for fungal pathogens encompass a range of approaches, including the use of resistant varieties, environmental sanitation, application of chemical pesticides, and utilization of fungal antagonists. While chemical pesticides offer short-term efficacy, their long-term use is associated with detrimental consequences, such as the development of pathogen resistance and environmental toxicity [3,10,11]. This has led to a push for more environmentally friendly options, such as biological control agents, which offer effective control with minimal side effects. Over the years, there has been significant progress in the development of several biological control agents, particularly fungal antagonists like *Trichoderma* spp. and *Bacillus subtilis*, for fungal control in soybean and other crop production [12,13,14]. However, while these organisms show promising results in controlled conditions, their field efficacy can be impacted by environmental conditions, including climate change, which influences their biological processes [15]. To combat this, science must accelerate the adaptation of these bioagents to changing climatic conditions by selecting agents better suited to the new climate. This underscores the importance of not only improving our methods for combating resistant pathogens but also developing new antifungal agents to be integrated into current control measures.

Ecosystems are characterized by diverse life forms, from microscopic viruses to complex organisms, engaged in intricate relationships that range from mutually beneficial partnerships to exploitative parasitism [16,17]. These symbiotic associations offer significant potential for addressing challenges across various fields, particularly in disease treatment, where they can inspire the development of alternatives to synthetic chemicals [18,19]. A key aspect of these interactions is the production of specialized metabolites by microbes, which mediate their interactions with other organisms. These molecules function as defensive or offensive weapons in microbial competition and defense. Notably, several of these natural products (NPs) show promise as therapeutic compounds or templates for designing novel structural scaffolds [18,19].

The bacterial genera *Photorhabdus* and *Xenorhabdus* are established sources of novel natural products that have antifungal, antibacterial, or insecticidal properties [20,21]. These bacteria are associated with entomopathogenic *Heterorhabditis* and *Steinernema* spp. nematodes, found in soil and have a significant potential as biological control agents of agricultural and public health pests [22,23]. This nematode–bacteria symbiosis targets the hemolymph of insect hosts, resulting in the production of a diverse array of secondary metabolites. These metabolites play crucial roles in suppressing insect immunity, defending against competitors, and deterring natural enemies [24,25]. Currently, *Xenorhabdus* is classified into 32 distinct taxa, with 31 recognized species and one species divided into two subspecies. Similarly, *Photorhabdus* encompasses 30 taxa, consisting of 23 species and six species that are further subdivided into subspecies [26]. Numerous studies have demonstrated the antifungal activity of metabolites in supernatants or extracts produced by these bacteria against various phytopathogenic fungi [27,28,29,30,31,32,33,34,35,36,37]. Genomes of these bacteria contain an impressive number of biosynthetic gene clusters (BGCs) associated with specialized metabolite production [38,39] and mainly encoding non-ribosomal peptide synthetases (NRPS) and hybrids of NRPS and polyketide synthases (PKS) [40]. Several bioactive compounds with antifungal properties have been identified, including xenocoumacin 1 (Xcn1), cabanillasin, and fabclavines from *Xenorhabdus* species [41,42,43], and *trans*-cinnamic acid (TCA) and benzaldehyde from *Photorhabdus* species [27,44]. While research has focused on a few species, exploring the metabolic potential of the remaining species and subspecies within both genera could reveal novel bioactive compounds.

This study aimed to (i) evaluate inhibitory effects of cell-free supernatants (CFSs) from four symbiotic bacteria, including a local isolate (*X. szentirmaii* PAM 25), against four phytopathogenic fungi; (ii) identify the antifungal bioactive compounds produced by mutant strains of *X. szentirmaii* and *X. doucetiae*, which were generated by easyPACId (Easy Promoter Activated Compound Identification) approach; (iii) evaluate the shelf stability of *X. szentirmaii* PAM 25 bacterial liquid culture and its secondary metabolites; and (iv) assess the efficiency of *X. szentirmaii* PAM 25 liquid culture and its metabolites in protecting soybean seeds (BMX Potência cultivar) against *Sclerotinia sclerotiorum*. This study expands upon previous work by combining the evaluation of various bacterial strains, identifying specific bioactive compounds, and assessing their shelf stability and seed treatment efficacy.

## 2. Materials and Methods

### 2.1. Maintenance of Phytopathogenic Fungal Cultures

*Fusarium oxysporum*, *Botrytis cinerea*, and *Macrophomina phaseolina* fungal isolates were sourced from the Department of Plant Protection, Faculty of Agriculture at Aydin Adnan Menderes University, Türkiye. *Sclerotinia sclerotiorum* was isolated from soybean plants cultivated in Pilar do Sul, Sao Paulo, SP, Brazil, and was obtained from the Mycological Collection of the Phytopathology Laboratory at the Instituto Biológico, Campinas-SP, Brazil. *Fusarium oxysporum*, *B. cinerea*, and *M. phaseolina* were incubated at 25 °C in the dark for 7–14 days prior to use, while *S. sclerotiorum* was kept at 22 °C.

### 2.2. Xenorhabdus spp. and Photorhabdus spp. Sources

The wildtype bacterial strains of *Xenorhabdus* spp. and *Photorhabdus* spp. were obtained from the American Type Culture Collection or the German Collection of Microorganisms and Cell Culture. These reference strains were sourced from the Invertebrate Animals Laboratory’s microorganism collection at Aydin Adnan Menderes University, Türkiye (Appendix A). *Xenorhabdus szentirmaii* PAM 25, a Brazilian isolate, was provided by the Biological Pest Control Laboratory at the Instituto Biológico-CAPSA, Campinas-SP, Brazil. All bacterial strains were stored in 50% glycerol solution at −80 °C.

### 2.3. Preparation of Bacterial Supernatants

The strains of *Xenorhabdus* spp. and *Photorhabdus* spp. were subcultured from stock cultures on Luria–Bertani (LB) agar (tryptone 10 g/L, yeast extract 5 g/L, NaCl 5 g/L, agar 15 g/L) and incubated in a BOD incubator at 28 °C for 24 h. Subsequently, a single colony of the respective strain was transferred to a 100 mL Erlenmeyer flask containing 10 mL of LB Broth and incubated at 30 °C for 24 h under constant agitation at 150 rpm. After this period, the cell concentration was determined using a spectrophotometer (OD_600_ = 1), and the phase I status of the bacteria was confirmed through isolation on NBTA medium (nutrient agar, bromothymol blue, triphenyl tetrazolium chloride) or the catalase test [45]. Five mL samples of the bacterial cultures were inoculated into a 1 L Erlenmeyer flask containing 500 mL of LB medium (1 mL of culture per 100 mL of medium) and incubated at 30 °C for 6 days under constant agitation at 150 rpm [46,47]. After this period, the bacterial culture was transferred to 50 mL Falcon tubes, and the supernatant was obtained by centrifugation (10,000 rpm, 4 °C, 10 min.). The supernatant was filtered through a 0.22 µm cellulose filter and stored at −20 °C for up to two weeks prior to use in the experiments [48].

### 2.4. Antifungal Activity of Cell-Free Supernatants of Wildtype Xenorhabdus spp. and Photorhabdus spp.

Obtained cell-free supernatants (CFS) were incorporated into potato dextrose agar (PDA) at 5, 10, and 20% (*v*/*v*) concentrations. The bacterial supernatants were added after the autoclaved PDA medium was allowed to cool to 45–50 °C, and the solution was thoroughly mixed before pouring the plates. The positive control consisted of PDA medium plus the fungicide Maxim^®^XL (active ingredients: Metalaxyl-M + Fludioxonil) at a ratio of 1 mL fungicide per 100 mL medium. In the negative control group, no supernatant was added; instead, sterile LB medium was added to the Petri dishes. The fungal phytopathogens were added to the prepared plates by transferring a 5 mm diameter mycelial plug from an actively growing fungal culture using a sterile transfer tube onto the center of each Petri dish containing bacterial supernatant or LB (control) [29,41]. Plates were incubated at 25 °C in darkness, and two perpendicular diameter measurements of fungal growth were taken after 14 days of incubation, and the average was calculated. The colony area was calculated as πr^2^. The area of the 5 mm plug in the middle of the plate was not included in the measurement [33]. Each treatment (4 fungi, 5 bacteria, and 3 supernatant concentrations) comprised 10 Petri dishes, and experiments were conducted in triplicate on different dates.

### 2.5. Determination of Antifungal Bioactive Compounds Produced by X. szentirmaii and X. doucetiae

The generation of mutants was described previously [49]. The strains, maintained at the Max Planck Institute for Terrestrial Microbiology in Marburg, Germany, were used in this study, with a total of 10 strains evaluated (Table 1).

### 2.6. Determination of Bioactive Compounds Using easyPACId Strains

The antifungal bioactive compounds of *X. szentirmaii* and *X. doucetiae* easyPACId strains were evaluated against the mycelial growth of *S. sclerotiorum*, *B. cinerea*, *M. phaseolina*, and *F. oxysporum*. Initially, the mutant strains were cultured in LB agar medium supplemented with 50 μg/mL of kanamycin (final concentration), while the WT, Δ*hfq,* and Δ*pptase* control strains were grown in LB medium without kanamycin [50]. All strains were incubated at 30 °C for 48 h under agitation at 150 rpm. For the pre-culture, a single colony of each strain was inoculated into 10 mL of LB medium + kanamycin and incubated at 30 °C under constant agitation at 150 rpm for 24 h. Subsequently, the pre-cultures were inoculated into 20 mL of LB medium with an optical density (OD_600_) adjusted to 0.1 and incubated at 30 °C for 1 h. Afterwards, the cultures were induced with 0.2% L-arabinose and incubated for 72 h at 30 °C under agitation at 150 rpm [46,50]. Subsequently, the bacterial culture was transferred to 50 mL Falcon tubes and centrifuged at 10,000 rpm, 4 °C for 10 min. The supernatant was filtered through a 0.22 µm cellulose filter and stored at −20 °C for up to 2 weeks prior to use in the experiments [48].

Cell-free supernatants from 10 strains (Table 2) were prepared and incorporated into PDA medium at a concentration of 20%. The experimental design and number of replicates per treatment were identical to the previous assay. The controls included the wildtype *X. szentirmaii* and *X. doucetiae,* as well as the respective Δhfq strains, while the negative control was PDA medium. Each treatment had 10 replicates, and the experiments were conducted three times.

### 2.7. Shelf Stability of X. szentirmaii Secondary Metabolites

The study assessed the shelf stability of *X. szentirmaii* PAM 25 secondary metabolites. This is a crucial point for developing practical applications, as stability is essential for commercial products. *X*. *szentirmaii* PAM 25 was grown in LB medium, maintained at 27 °C for 6 days under constant agitation at 150 rpm. Afterwards, the culture was centrifuged to obtain cell-free supernatants (CFS), as described previously.

A 25 mL aliquot of CFS, including secondary metabolites, was transferred to separate 100 mL Schott bottles and stored at three different temperatures: 5 °C, 25 °C, and 35 °C. The contents of the bottles were evaluated at the following time points: 0, 7, 15, 30, 60, 90, 120, 150, 180, 210, 240, 270, 300, 330, and 360 days. One bottle was prepared for each temperature and storage time. At each time point, the CFS from each temperature was added to the PDA medium at a 10% ratio. For control, only LB medium was added. A 5 mm diameter disk of *S. sclerotiorum* mycelial culture was inoculated in the center of each plate. Sixteen replicates were established for each treatment, each replicate represented by a 100 mL Schott bottle.

The evaluated parameter was fungal inhibition, measured as the colony diameter (cm) in the medium until the control treatment (PDA) reached the full diameter of the plate. The experiment was concluded when the metabolites caused ≥50% inhibition of the mycelial growth.

The percentage inhibition of *S. sclerotiorum* mycelial growth for each treatment was compared to the control treatment using the following formula: (Dc − Dt)/Dc × 100%, where Dc and Dt represent the colony diameters of the control and treatment, respectively.

### 2.8. Efficacy of X. szentirmaii Broth Culture and Its Cell-Free Supernatant in Soybean Seed Protection

This study evaluated the efficiency of *X. szentirmaii* PAM 25 metabolites in protecting soybean seeds (BMX Potência cultivar) against *S. sclerotiorum*. This aims to demonstrate the potential of these bacteria and their secondary metabolites for agricultural applications.

*Xenorhabdus szentirmaii* PAM 25, which showed the highest inhibition rate in the first experiment, was cultivated in an Erlenmeyer flask containing 300 mL of LB medium, maintained at 27 °C for 6 days under constant agitation at 150 rpm. The bacterial broth culture was divided into two parts, and one part was centrifuged at 10,000 rpm for 10 min and filtered to obtain cell-free supernatant, as described in the previous experiment.

Six treatments were established, including the following:

Cell-free Supernatant at 10%;

Bacterial liquid culture at 10%;

LB medium (negative control);

Distilled water (negative control);

CERTEZA^®^ fungicide (a.i. Tiafanato metilico + fluazinam; dose 56.0 µL/28.0 g of seeds) (positive control);

TrichoTurbo^®^ biofungicide (a.i. *Trichoderma asperellum* BV10; dose 4 mL/kg of seeds) (positive control).

For adherence to the seeds, treatments containing bacterial broth culture or CFS and TrichoTurbo^®^ were supplemented with 1% xanthan gum.

The soybean seeds superficially disinfected with 2% sodium hypochlorite were placed in an autoclaved plastic bag. After adding the treatment materials, the bags were shaken vigorously for 3 min. The treated 10 seeds were placed in Petri dishes (90 × 15 mm) containing filter paper moistened with autoclaved distilled water. For treatments involving the phytopathogenic fungus, a 5 mm diameter disk of *S. sclerotiorum* mycelial culture (grown for 10 days on PDA at 20 °C in darkness) was placed over the hilum of the treated seeds. The Petri dishes were sealed with PVC film and maintained at 25 °C in darkness for 7–14 days. Each treatment had four replicates, with each replicate represented by one Petri dish. This experiment was conducted twice.

Seeds infected with *Sclerotinia* do not germinate or germinate poorly compared to healthy seeds. The evaluation consisted of counting the number of germinated seeds 10 days after the experiment setup. The data were expressed as the percentage of germinated seeds.

### 2.9. Data Analysis

The inhibition percentage data were transformed using the arcsin √(x/100) function, where x represents the percentage of each treatment’s replicate. The data on mycelial growth area were analyzed using ANOVA. The main factors taken into consideration were the different bacterial strains (5 total), the four fungal pathogens, and the three supernatant concentrations (5%, 10%, and 20%). Treatment means were compared using Tukey’s test at a 5% significance level. Statistical analyses were performed using SPSS version 23 and SISVAR DEX/UFLA, version 5.6.

## 3. Results

### 3.1. Antifungal Activity of Cell-Free Supernatants of Wildtype Xenorhabdus spp. and Photorhabdus spp.

A dose-dependent effect of cell-free supernatants of *Xenorhabdus* spp. and *Photorhabdus* spp. was observed against four phytopathogenic fungi. Across all tested fungi except *Fusarium*, the fungicide treatment consistently resulted in the highest mycelial inhibition, approaching 100%. However, the level of inhibition induced by the bacterial supernatants varied significantly among the different fungal species.

Against *S. sclerotiorum*, at 5% supernatant concentration, *X. szentirmaii* PAM 25 and DSM 16338 strains demonstrated the highest inhibition rates of 25.28 and 30.54%, respectively, whereas *P. kayaii* and *X. nematophila* exhibited the least efficacy (*p* < 0.05). There was a significant difference among the bacterial species (*F* = 317.62; df = 6, 133; *p* < 0.0001) (Figure 1A). The percentage of mycelial inhibition increased as the concentration of the supernatant increased from 5% to 20%. At 20%, *X. szentirmaii* DSM 16338 and PAM 25 show the most promising antifungal activity, with nearly 100% inhibition at 20% concentration, followed by *X. doucetiae* with 87% inhibition (Figure 1B). There were no significant differences among these three bacterial strains. In contrast, *P. kayaii* and *X. nematophila* presented relatively lower inhibition, ranging between 24 and 27% (F = 559.14; df = 6, 133; *p* < 0.0001) (Figure 1A).

Similarly, at 5% concentration, *X. szentirmaii* DSM 16338 showed a significant *B. cinerea* mycelial inhibition of 71.89%, followed by PAM 25 with 56.37% (*F* = 942.29; df = 6, 133; *p* < 0.0001) (Figure 2A). *Xenorhabdus nematophila* and *P. kayaii* exhibited considerably lower mycelial inhibition against *B. cinerea* across all tested concentrations compared to the other bacterial supernatants (Figure 2B). Among the bacterial supernatants at a 10% concentration, *X. szentirmaii* DSM 16338 exhibited the highest mycelial inhibition at 94.32%, followed by *X. doucetiae,* which demonstrated significant inhibition at 83.52%. *Xenorhabdus szentirmaii* PAM 25 strain showed a moderate level of inhibition at 69.90% (*F* = 1227.48; df = 6, 133; *p* < 0.0001). At 20%, both *X. szentirmaii* strains and *X. doucetiae* demonstrated the highest inhibition among the bacterial supernatants, reaching over 96.53% against *B. cinerea*. *Photorhabdus kayaii* showed the lowest inhibition, with only 16.82% at the highest concentration (20%) (*F* = 1123.13; df = 6, 133; *p* < 0.0001) (Figure 2A).

*Macrophomina phaseolina* demonstrated the highest susceptibility to the bacterial treatments. Mycelial inhibition induced by the bacterial supernatants generally increased with increasing supernatant concentration (Figure 3A,B). At a 5% concentration, *X. szentirmaii* DSM 16338 and PAM 25 strains demonstrated 88.19% and 71.83% inhibition, respectively. *Xenorhabdus nematophila* and *X. doucetiae* showed limited antifungal activity. There was a significant difference among the bacterial species (*F* = 1245.23; df = 6, 133; *p* < 0.0001) (Figure 3A). *Xenorhabdus doucetiae* showed moderate inhibition (71.16%) at 10% and high inhibition (94.83%) at 20%. In contrast, *P. kayaii* and *X. nematophila* displayed significantly lower mycelial inhibition across all tested concentrations compared to the other tested bacteria (*p* < 0.05). At 20%, *X. szentirmaii* DSM 16338 and *X*. *doucetiae* were the most effective, achieving high levels of inhibition (over 90%), followed by *X. szentirmaii* PAM 25 strain with 86% inhibition (*F* = 269.75; df = 6, 133; *p* < 0.0001) (Figure 3A).

*Fusarium oxysporum* generally showed lower levels of inhibition. *Xenorhabdus szentirmaii* DSM 16338 was the most effective bacterial supernatant at lower concentrations (5% and 10%), exhibiting mycelial inhibition of 52.43% and 64.66%, respectively (Figure 4A,B). Following this, *X. szentirmaii* PAM presented 33–48% fungal inhibition at lower concentrations (5% and 10%). There was a significant difference among the bacterial species (at 5% *F* = 429.50; df = 6, 133; *p* < 0.0001 and at 10% *F* = 346.32; df = 6, 133; *p* < 0.0001) (Figure 4A). At 20% (*F* = 204.06; df = 6, 133; *p* < 0.0001), *X. szentirmaii* DSM 16338 efficacy decreased to 41.32%, whereas *X. szentirmaii* PAM 25 demonstrated the highest inhibition (59.14%) among the bacterial treatments, even surpassing the fungicide’s effect (37.42%). *Xenorhabdus doucetiae* displayed relatively low inhibition at 5% (3.99%), which gradually increased to moderate levels at 10% (7.38%) and 20% (12.19%). In contrast, *X. nematophila* and *P. kayaii* exhibited minimal inhibitory effects against *F. oxysporum* across all tested concentrations (Figure 4A,B).

Overall, *F. oxysporum* was the most resistant to the treatments, followed by *S. sclerotiorum* and then *B. cinerea*. *M. phaseolina* was the most sensitive. (*F* = 1017.83; df = 3; *p* < 0.001)

### 3.2. Antifungal Compounds Produced by Xenorhabdus szentirmaii and X. doucetiae

Among the eight compounds generated by *X. szentirmaii* mutants, CFS of Δ*hfq* LP-56 mutants producing fabclavine provided the highest inhibition of the three fungi *S. sclerotiorum* (*F* = 572.92; df = 12, 259; *p* < 0.0001) (Figure 5A), *B. cinerea* (F = 203.07; df = 12, 259; *p* < 0.0001) (Figure 5B), and *M. phaseolina* (*F* = 5187.99; df = 11,239, *p*< 0.0001) (Figure 5C), with >90% inhibition. Antifungal inhibition rate of this CFS against *F. oxysporum* was 75.6% (*F* =1153.92; df = 12, 259; *p* < 0.0001) (Figure 5D). There was no significant difference between the CFS of wildtype and Δ*hfq* LP-56 mutants against *S. sclerotiorum* and *B. cinerea*. Fabclavine was the sole metabolite to significantly inhibit *M. phaseolina* and *F. oxysporum*, achieving over 70.0% inhibition, which significantly differed from the positive control (WT *X. szentirmaii* DSM 16338) (*p* < 0.001) (Figure 5A,B). Against *M. phaseolina* and *F. oxysporum*, most compounds showed limited inhibitory activity, with significant inhibition observed only for the positive control (*X. szentirmaii* DSM 16338) and fabclavine (Figure 5C,D).

In addition to fabclavine, pyrollizixenamide and szentirazine inhibited *S. sclerotiorum* mycelial growth by 60.3% and 28.8%, respectively (Figure 5); whereas rhabdopeptid, pyrollizixenamide, and xenobactin inhibited *B. cinerea* mycelial growth by less than 50%, significantly differing from the other compounds (*p* < 0.0001) but not from each other (*p* = 0.372).

For *X. doucetiae*, four secondary metabolites were tested. Xenocoumacins significantly inhibited *S. sclerotiorum* (*F* = 173.988; df = 5, 119; *p* < 0.0001) and *M. phaseolina* (*F* = 740.804; df = 5, 119; *p* < 0.0001), achieving inhibition rates above 85.0%. This inhibition differed significantly from that of the other compounds but not from the positive control (*X. doucetiae* wildtype) (Figure 6A,C). All tested SMs produced by *X. doucetiae* (xenorhabdin, phenylethylamides, tryptamides, and xenocoumacin) inhibited *B. cinerea* mycelial growth by 38.7% to 59.1%, significantly lower than the *X. doucetiae* wildtype, which inhibited 98.5% (*F* = 46.956; df = 5, 119; *p* < 0.0001) (Figure 6B). Against *F. oxysporum*, all tested compounds showed minimal inhibitory activity, ranging from 0.35% to 1.02%, whereas *X. doucetiae* wildtype inhibited only 8.7%, significantly differing from the *X. doucetiae* mutants (F = 27.555; df = 5, 119; *p* < 0.0001) (Figure 6D).

### 3.3. Shelf Stability of Xenorhabdus szentirmaii Secondary Metabolites

*Xenorhabdus szentirmaii* PAM 25 supernatant provided >80% inhibition of *S. sclerotiorum* growth throughout the storage period (360 days) at all three tested temperatures (Table 2).

### 3.4. Efficacy of Xenorhabdus szentirmaii Broth Culture and Its Cell-Free Supernatant for Soybean Seed Protection

The broth culture and cell-free supernatant of *X. szentirmaii* PAM 25, both diluted at 10% and undiluted, were effective in inhibiting *S. sclerotiorum* in seed treatments, ensuring ≥80% seed germination, with no significant difference compared to the fungicide treatment (thiophanate-methyl + fluazinam) (Figure 7). Between 40 and 54% of soybean seeds in the *Trichoderma asperellum-treated group* germinated, demonstrating *Trichoderma*’s low efficacy against *S. sclerotiorum* in seed protection and its negative effect on seed viability. There was a significant difference between the treatments (*F* = 4.486; df = 7, 56; *p* < 0.0005) (Figure 7).

## 4. Discussion

This study evaluated the antifungal potential of cell-free supernatants from four symbiotic bacteria, including a local isolate, *X. szentirmaii* PAM 25, against four phytopathogenic fungi. Furthermore, it assessed the efficacy of *X. szentirmaii* PAM 25 and its metabolites in protecting soybean seeds from *S. sclerotiorum* and investigated the shelf stability and seed treatment effects of these bacterial metabolites.

Among the isolates tested, *X. szentirmaii* DSM 16338, *X. szentirmaii* PAM 25, and *X. doucetiae* demonstrated significant inhibition against *S. sclerotiorum*, *B. cinerea*, and *M. phaseolina*, exhibiting superior efficacy compared to *X. nematophila* and *P. kayaii*. This is consistent with numerous reports highlighting the antifungal activity of bacterial metabolites against various phytopathogens. Cell-free supernatants from bacteria species, including *X. khoisanae*, *X. nematophila*, *X. szentirmaii*, *P. kayaii*, *P. laumondii*, and *P. temperata,* have shown substantial inhibition of mycelial growth in a wide range of fungi, such as *Fusicladium carpophilum*, *F. effusum*, *Armillaria tabescens*, *B. cinerea*, *Fusarium oxysporum*, *F. solani*, *Phytophthora capsica*, *Rhizoctonia solani*, *S. sclerotiorum*, and *Neofusicoccum parvum*, *Colletotrichum gloeosporioides* [27,28,29,30,31,34,35,36,37,51]. Generally, *Xenorhabdus* species exhibited greater antifungal activity compared to *Photorhabdus* species, and fungal susceptibility to bacterial metabolites varied. *Botrytis cinerea* proved particularly susceptible to *X. szentirmaii* DSM 16338. This high inhibition rate of *B. cinerea* by *Xenorhabdus* species is consistent with previous studies [29,30,52]. However, other bacterial species (*P. kayaii* and *X. nematophila)* were far less effective against *B. cinerea.* In contrast, Fang et al. [25] reported high efficacy of *X. nematophila* supernatant against *B. cinerea* and *P. capsici* in vitro at 10% supernatant concentration. This difference may be due to the different isolates, both *Xenorhabdus* species and isolates of fungal pathogens, used in the studies.

The varying susceptibility among fungi is a key finding. For example, *Fusarium oxysporum* demonstrated greater resistance to the bacterial supernatant compared to the other fungal pathogens. The varying levels of inhibition by *X. szentirmaii* supernatant, depending on the strain and concentration, align with previous studies [53] that also highlight this pathogen’s resistance to *Xenorhabdus* metabolites. This is due to its high adaptability, which is not limited to genetic diversity. Its robust cell wall, composed of complex layers of chitin and glucans, acts as a formidable barrier, while its ability to produce detoxification enzymes helps it neutralize foreign compounds. This resistance poses a significant challenge for sustainable disease management [54,55]. To overcome it, future strategies could involve combining different bacterial metabolites to create a synergistic effect, targeting multiple defense mechanisms simultaneously. While *F. oxysporum* showed resistance, Sharma et al. [56] demonstrated that *Xenorhabdus assam-isolate* (Sg as1) metabolites, extracted with ethyl acetate and applied at 1000 µg/mL, exhibited significant antifungal activity against *M. phaseolina* (EC_50_ = 55.98 µg/mL), resulting in 70% mycelial inhibition. These results, along with the numerous reports of inhibition against a wide range of fungi, underscore the potential of *Xenorhabdus* and *Photorhabdus* as biocontrol agents while also emphasizing the need for a targeted approach.

Following the evaluation of cell-free supernatants, this study further sought to identify the specific antifungal bioactive compounds produced by mutant strains of *X. szentirmaii* and *X. doucetiae*, which were generated using the easyPACId approach. Fabclavines from *X. szentirmaii* showed broad-spectrum activity against all four tested fungi. Xenocoumacin from *X. doucetiae* inhibited *S. sclerotiorum* and *M. phaseolina* but was ineffective against *F. oxysporum*. Both fabclavines and xenocoumacins are hybrid hexapeptide/polyketide water-soluble peptides synthesized by non-ribosomal peptide synthetases (NRPS) and polyketide synthases (PKS) enzyme systems [44,55,57]. These compounds have various biological activities, including antibacterial and antifungal activities, playing key roles in inhibiting saprophytic and competitive organisms from infecting insect hosts [58]. The complex structures of fabclavines and xenocoumacins suggest that they may interact with multiple cellular targets, contributing to their observed antifungal activity. While precise mechanisms are still under investigation, potential pathways include disruption of fungal cell membrane integrity, leading to leakage and interference with essential transport processes for fabclavines [57,58], and inhibition of protein translation and cell wall component synthesis for xenocoumacins [59,60]. The leakage of vital intracellular components after disrupting cell membrane structure could explain fabclavines broad-spectrum efficacy against pathogens such as *S. sclerotiorum* and *Botrytis cinerea*. Different *Xenorhabdus* species produce a range of fabclavine and xenocoumacin derivatives, exhibiting substantial structural and bioactivity variations [57,58]. *Xenorhabdus doucetiae* does not harbor the biosynthesis gene cluster involved in fabclavine production [58]. Beyond their antifungal activity, both fabclavine and xenocoumacin have been established to have great potential in pathogen treatment and pest control [39,43,46,61,62]. Sharma et al. [63] reported strong antifungal activity from ethyl acetate extracts of *X. assam* (Sg as1) filtrates against *M. phaseolina*, detecting xenocoumacin, xenorhabdin, and nematophin as the three main antifungal compounds. In this study, the three compounds were evaluated separately, with xenocoumacin exhibiting the highest antifungal activity.

Pyrollizixenamide and szentirazine produced by *X. szentirmaii* also exhibited moderate antifungal effects against *S. sclerotiorum* and *B. cinerea*, ranging between 23.8 and 60.3%. The other tested compounds showed limited inhibitory activity against these pathogens. Pyrollizixenamide and szentirazine are part of the complex chemical arsenal that *Xenorhabdus* bacteria use to interact with their environment [49,64]. Their specific properties and mechanisms remain largely uncharacterized.

This study also investigated the shelf stability and seed treatment effects of these bacterial metabolites. The cell-free supernatant of *X. szentirmaii* PAM 25 exhibited >94.0% inhibition of *S. sclerotiorum* mycelial growth after one year of storage at 25 °C. The bacterial culture and supernatants of *X. szentirmaii* PAM 25 effectively control *S. sclerotiorum* in soybean seeds, allowing for more than 80% seed germination, demonstrating similar efficacy as the thiophanate-methyl + fluazinam fungicide. This bacterium was more effective than the fungal biocontrol product, *T. asperellum,* which exhibited low efficacy against *S. sclerotiorum* in seed protection and had a negative effect on seed viability. Similarly, the antifungal activity of *X. szentirmaii* bacterial culture and supernatants against *Monilinia fructicola* remained stable for 9 months when stored at 20 °C and 4 °C [65]. In the case of seed treatments, undiluted and 10% bacterial broth culture and supernatants of *X. szentirmaii* PAM 25 ensured >80% soybean seed germination in the presence of *S. sclerotiorum*. In a previous study, a 33% diluted *X. szentirmaii* PAM 25 bacterial culture effectively inhibited *S. sclerotiorum* on soybean seeds and promoted significant seed germination and plant development [28].

This study highlights the potential of *Xenorhabdus* species, particularly *X. szentirmaii*, as effective biological control agents. The identification of fabclavines and xenocoumacins as key antifungal compounds provides valuable insights into the mechanisms of action. Further research could delve deeper into their precise mechanisms of action. Investigating the molecular interactions between these compounds and fungal cells could reveal novel targets for antifungal drug development. The demonstration of shelf stability and seed treatment efficacy underscores the feasibility of direct seed application of *Xenorhabdus* bacteria. This practical approach offers a safe and novel biological control method, presenting a promising alternative to environmentally harmful chemical fungicides. While the potential is clear, three key hurdles must be addressed to transform these metabolites into a commercially viable biofungicide. The high cost and complexity of producing and purifying bacterial metabolites must be overcome. For the product to be cost-competitive with chemical alternatives, large-scale production must be scalable and economically feasible. A stable formulation is also essential. The raw metabolites must be combined with adjuvants and other inert ingredients to ensure the final product has a long shelf-life and is stable for effective field application. The most significant barrier is the rigorous, multi-year, and multi-million-dollar process of gaining regulatory approval. This requires extensive toxicological and environmental studies to prove the product’s safety for people and the ecosystem. In addition, in the field, the effectiveness of *Xenorhabdus* and *Photorhabdus* can be limited by how soil conditions affect bacterial viability, competition with the native soil microbiome, and the degradation of its antifungal compounds by environmental factors like UV radiation and rainfall. Future research should focus on conducting field trials to evaluate the efficacy of *Xenorhabdus* and *Photorhabdus* bacteria and their metabolites under real-world agricultural conditions, as well as investigating their effects on non-target organisms and soil health.

## Figures and Tables

**Figure 1 jof-11-00691-f001:**
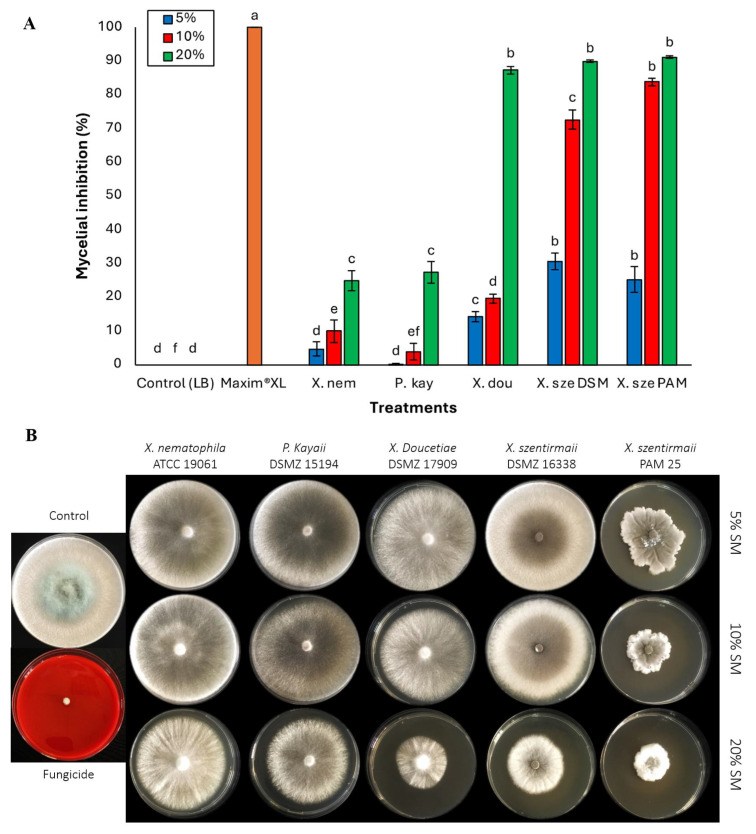
(**A**) Activities of cell-free supernatants at concentrations of 5%, 10%, and 20% on phytopathogenic fungi *Sclerotinia sclerotiorum*. Data on mycelial growth area presented as mean mycelial inhibition ± standard error. The effects of each concentration were analyzed using ANOVA with Tukey’s test (*p* < 0.05). Different letters above the bars represent statistical differences at the tested concentrations. Abbreviations: Maxim^®^XL (fungicide), X. nem: *Xenorhabdus nematophila*, P. kay: *Photorhabdus kayaii*, X. dou: *Xenorhabdus doucetiae*, X. sze DSM: *Xenorhabdus szentirmaii* DSM 16338, X. sze PAM: *Xenorhabdus szentirmaii* PAM 25. (**B**) Inhibition of mycelial growth of the fungal pathogen *Sclerotinia sclerotiorum* at concentrations of 5%, 10%, and 20% of the respective bacterial cell-free supernatant.

**Figure 2 jof-11-00691-f002:**
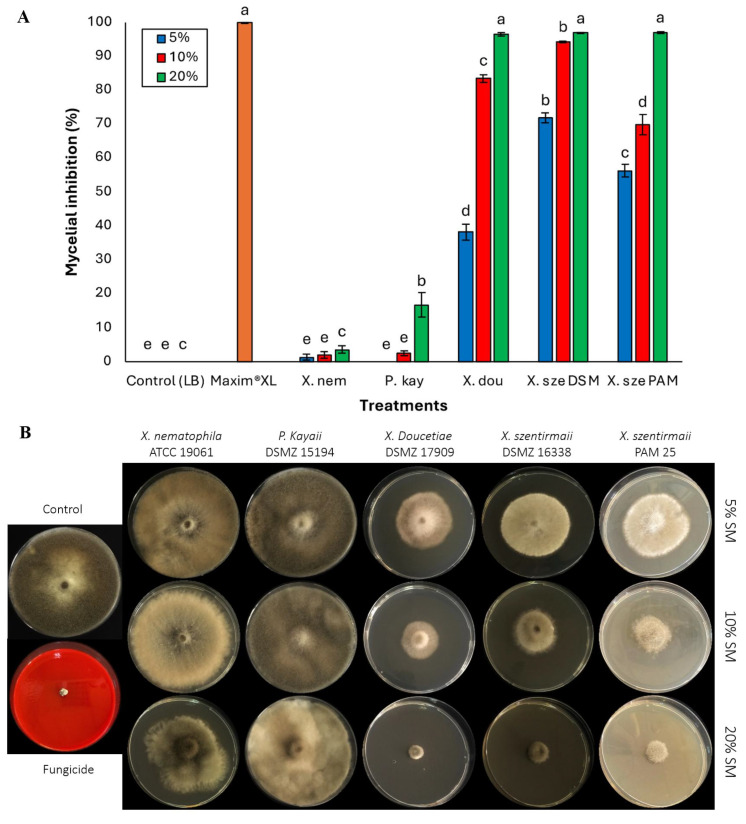
(**A**) Activities of cell-free supernatants at concentrations of 5%, 10%, and 20% on phytopathogenic fungi, *Botrytis cinerea*. Data on mycelial growth area presented as mean mycelial inhibition ± standard error. The effects of each concentration were analyzed using ANOVA with Tukey’s test (*p* < 0.05). Different letters above the bars represent statistical differences at the tested concentrations. Abbreviations: Maxim^®^XL (fungicide), X. nem: *Xenorhabdus nematophila*, P. kay: *Photorhabdus kayaii*, X. dou: *Xenorhabdus doucetiae*, X. sze DSM: *Xenorhabdus szentirmaii* DSM 16338, X. sze PAM: *Xenorhabdus szentirmaii* PAM 25. (**B**) Inhibition of mycelial growth of the fungal pathogen *Botrytis cinerea* at concentrations of 5%, 10%, and 20% of the respective bacterial cell-free supernatant.

**Figure 3 jof-11-00691-f003:**
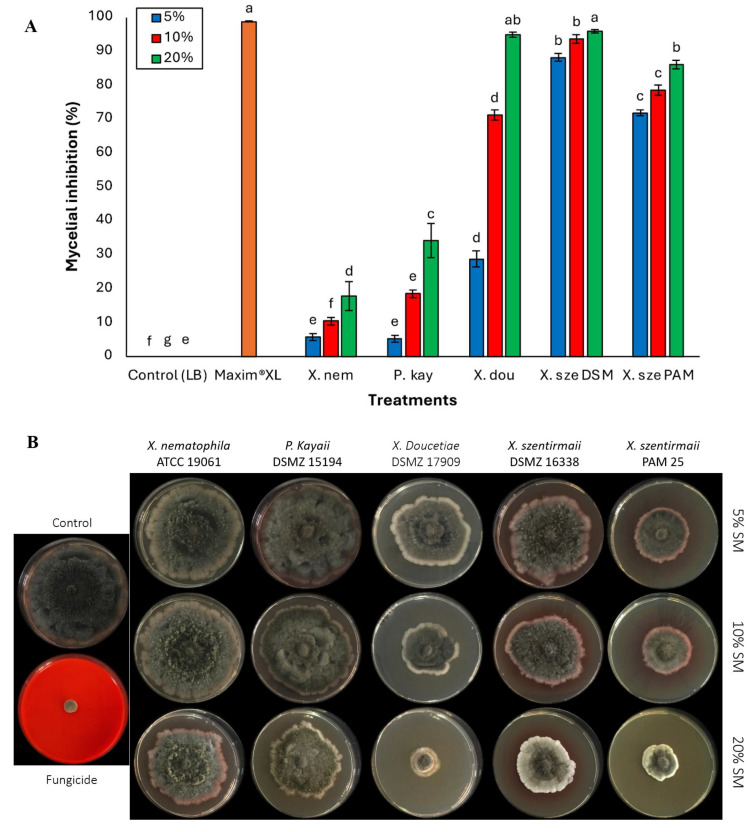
(**A**) Activities of cell-free supernatants at concentrations of 5%, 10%, and 20% on phytopathogenic fungi *Macrophomina phaseolina*. Data on mycelial growth area presented as mean mycelial inhibition ± standard error. The effects of each concentration were analyzed using ANOVA with Tukey’s test (*p* < 0.05). Different letters above the bars represent statistical differences at the tested concentrations. Abbreviations: Maxim^®^XL (fungicide), X. nem: *Xenorhabdus nematophila*, P. kay: *Photorhabdus kayaii*, X. dou: *Xenorhabdus doucetiae*, X. sze DSM: *Xenorhabdus szentirmaii* DSM 16338, X. sze PAM: *Xenorhabdus szentirmaii* PAM 25. (**B**) Inhibition of mycelial growth of the fungal pathogen *Macrophomina phaseolina*. at concentrations of 5%, 10%, and 20% of the respective bacterial cell-free supernatant.

**Figure 4 jof-11-00691-f004:**
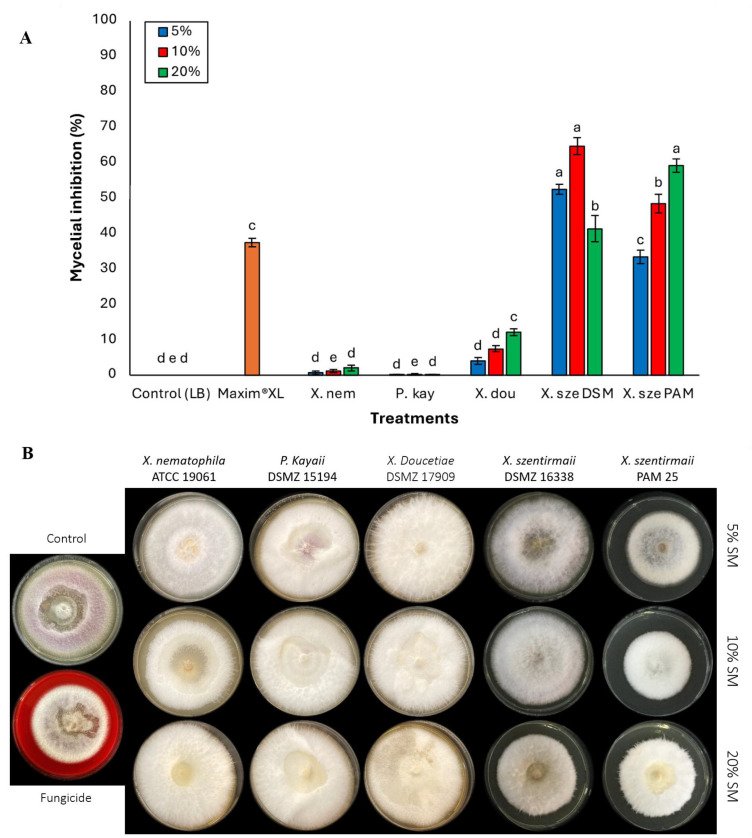
(**A**) Activities of cell-free supernatants at concentrations of 5%, 10%, and 20% on phytopathogenic fungi *Fusarium oxysporum*. Data on mycelial growth area presented as mean mycelial inhibition ± standard error. The effects of each concentration were analyzed using ANOVA with Tukey’s test (*p* < 0.05). Different letters above the bars represent statistical differences at the tested concentrations. Abbreviations: Maxim^®^XL (fungicide), X. nem: *Xenorhabdus nematophila*, P. kay: *Photorhabdus kayaii*, X. dou: *Xenorhabdus doucetiae*, X. sze DSM: *Xenorhabdus szentirmaii* DSM 16338, X. sze PAM: *Xenorhabdus szentirmaii* PAM 25. (**B**) Inhibition of mycelial growth of the fungal pathogen *Fusarium oxysporum*. at concentrations of 5%, 10%, and 20% of the respective bacterial cell-free supernatant.

**Figure 5 jof-11-00691-f005:**
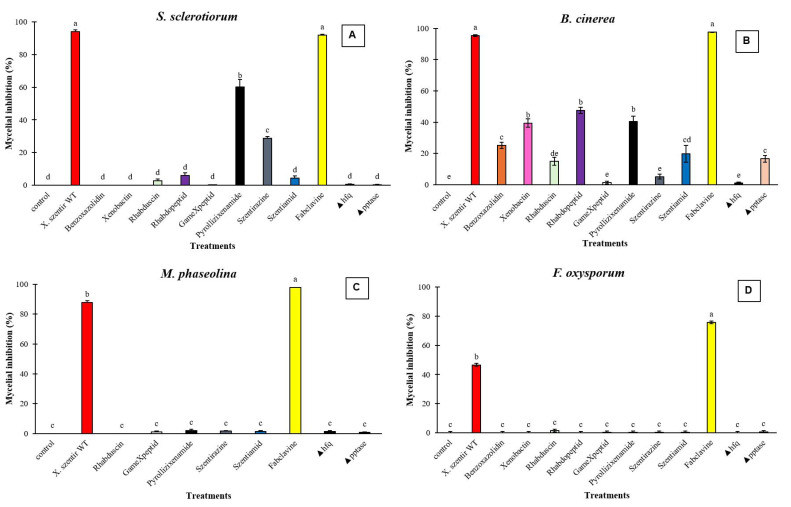
Activities of supernatants from each secondary metabolite produced by the respective easyPACId strain *Xenorhabdus szentirmaii* Δ*hfq* and Δ*pptase* mutants on phytopathogenic fungi. Negative control: LB agar. Positive control: *X. szentirmaii* DSM 16338 (wildtype) isolate. (**A**) *Sclerotinia sclerotiorum*, (**B**) *Botrytis cinerea*, (**C**) *Macrophomina phaseolina*, (**D**). *Fusarium oxysporum*. Data on mycelial growth area presented as mean mycelial inhibition ± standard error. The effects of each concentration were analyzed using ANOVA with Tukey’s test (*p* < 0.05). Different letters above the bars represent statistical differences at the tested concentrations.

**Figure 6 jof-11-00691-f006:**
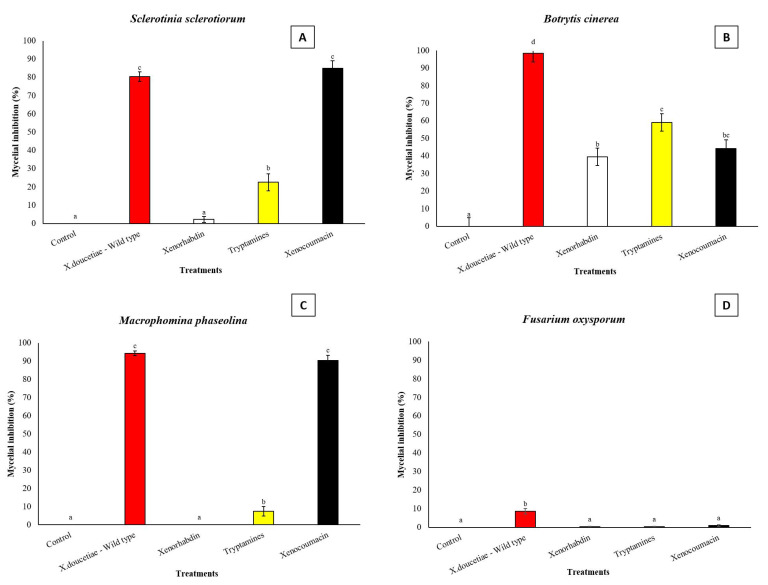
Activities of supernatants enriched with selected natural products produced by the respective easyPACId strains of *Xenorhabdus doucetiae* Δ*hfq* on phytopathogenic fungi. Negative control: LB agar. (**A**) *Sclerotinia sclerotiorum*, (**B**) *Botrytis cinerea*, (**C**) *Macrophomina phaseolina*, (**D**) *Fusarium oxysporum*. Data on mycelial growth area presented as mean mycelial inhibition ± standard error. The effects of each concentration were analyzed using ANOVA with Tukey’s test (*p* < 0.05). Different letters above the bars represent statistical differences at the tested concentrations.

**Figure 7 jof-11-00691-f007:**
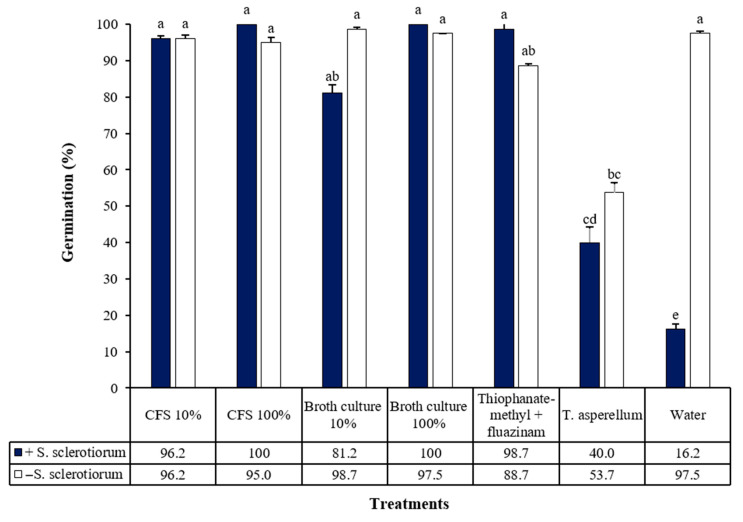
Percentage of soybean seed germination treated with *Xenorhabdus szentirmaii* PAM 25 culture broth and cell-free supernatant (CFS), diluted at different concentrations, followed by inoculation with the fungus *Sclerotinia sclerotiorum*. Identical lowercase letters indicate no significant differences between treatments, according to Tukey’s test at a 5% significance level. CV (%) = 8.5.

**Table 1 jof-11-00691-t001:** Δhfq mutants of *Xenorhabdus szentirmaii* and *X. doucetiae* generated using the easyPACid method and their synthesized molecules.

MUTANT STRAIN	SECONDARY METABOLITE
*X.szentirmaii* Δ*hfq* KS-16	GameXPeptide
*X.szentirmaii* Δ*hfq* KS-22	Pyrollizixenamide
*X.szentirmaii* Δ*hfq* LP-56	Fabclavine
*X.szentirmaii* Δ*hfq* SVS-204	Szentirazine
*X.szentirmaii* Δ*hfq* SVS-208	Szentiamide
*X.szentirmaii* Δ*hfq* SVS-212	Xenobactine
*X.szentirmaii* Δ*hfq* SVS-240	Rhabduscine
*X.szentirmaii* Δ*hfq* SVS-247	Rhabdopeptide
*X.szentirmaii* Δ*hfq*	-
*X.szentirmaii* Δ*pptase*	-
*X. doucetiae* HBLC-289	Xenorhabdin
*X. doucetiae* HBLC-107	Xenocoumacin
*X. doucetiae* HBLC-290	Phenylethylamides/Tryptamides

**Table 2 jof-11-00691-t002:** Percentage of mycelial growth inhibition of *Sclerotinia sclerotiorum* tested with supernatants of *Xenorhabdus szentirmaii* PAM 25 stored at different temperatures for up to 360 days.

Time (days)	Temperature (Mean ± SD)								
	5 °C			25 °C			35 °C		
0	99.30 ± 1.49	a ^1^	A ^2^	99.30 ± 1.49	a	A	99.30 ± 1.49	a	A
7	97.89 ± 1.72	ab	C	98.95 ± 1.46	a	AB	99.34 ± 1.39	a	A
15	94.84 ± 4.23	cd	B	96.68 ± 3.46	b	A	94.77 ± 2.97	cd	B
30	95.94 ± 1.59	c	A	94.77 ± 1.54	c	AB	93.90 ± 1.07	d	C
60	98.32 ± 1.64	a	A	98.55 ± 1.29	ab	A	96.13 ± 2.21	bc	B
90	97.77 ± 0.86	ab	A	98.13 ± 0.76	ab	A	94.84 ± 0.68	cd	B
120	99.18 ± 2.17	a	A	93.40 ± 2.47	c	B	93.36 ± 1.26	d	B
150	92.58 ± 1.68	e	B	94.45 ± 3.27	c	A	86.68 ± 7.28	f	C
210	98.55 ± 1.25	a	A	94.83 ± 1.61	c	B	88.16 ± 1.12	d	B
240	93.91 ± 1.73	de	A	83.59 ± 3.68	d	C	84.61 ± 0.65	e	B
270	94.42 ± 0.74	cde	A	94.12 ± 0.76	c	A	89.67 ± 0.56	bcd	B
300	97.97 ± 1.02	a	A	93.71 ± 0.21	c	AB	89.12 ± 0.64	cd	C
330	97.85 ± 1.40	ab	A	93.75 ± 0	c	B	91.07 ± 0.70	b	B
360	96.13 ± 1.63	bc	A	94.12 ± 0.76	c	AB	89.48 ± 0.61	bcd	C

^1^ Lowercase letters represent comparisons between storage days at each temperature (Tukey’s test, *p* < 0.05). ^2^ Uppercase letters represent temperature comparisons at each storage day (Tukey’s test, *p* < 0.05).

## Data Availability

All data generated or analyzed during this study are included in this published article. Any other information is available from the corresponding author on upon request.

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
