# Peer review of "Xenorhabdus and Photorhabdus Metabolites for Fungal Biocontrol and Application in Soybean Seed Protection"

_jof, 2025, doi:10.3390/jof11100691_

Round 1

Reviewer 1 Report

This study demonstrates that the direct seed application of Xenorhabdus and Photorhabdus bacteria offers a practical and innovative biological control method against soil-borne fungal pathogens. The experimental design is scientifically sound, and the manuscript is relatively well-structured. Nevertheless, several issues remain to be addressed, with the discussion section in particular requiring substantial refinement. 

Results

Line 118-119 Descriptions related to methods should be excluded from the Results section

Line 126 Italicize the letter P in the text

Line 128 Label the letter “A” on the top-left figure

It is suggested that the combination mode of the figures be modified:

Figure 1A+Figure 2/ Figure 1B+Figure 3/ Figure 1C+Figure 4/ Figure 1D+Figure 5

Line 235 Change Y-axis to “Mycelial inhibition (%)”

Line 211 Label the letter “ABCD” on the four figures

Line 252, 434 Reduce the font size in the chart

Discussion

The discussion section does not need to repeat the results. It is advisable to supplement the discussion content regarding this part of the results.

Author Response

Results

Line 118-119 Descriptions related to methods should be excluded from the Results section

Response: Deleted and there are no more descriptions related to methods in the results section.

Line 126 Italicize the letter P in the text

Response: Done

Line 128 Label the letter “A” on the top-left figure

Response: Done

It is suggested that the combination mode of the figures be modified:

Figure 1A+Figure 2/ Figure 1B+Figure 3/ Figure 1C+Figure 4/ Figure 1D+Figure 5

Response: Done

Line 235 Change Y-axis to “Mycelial inhibition (%)”

Response: Done

Line 211 Label the letter “ABCD” on the four figures

Response: Done

Line 252, 434 Reduce the font size in the chart

Response: Done

Discussion

The discussion section does not need to repeat the results. It is advisable to supplement the discussion content regarding this part of the results.

Response: Results repetitions have been removed from discussion

Reviewer 2 Report

This manuscript presents interesting data to show the biocontrol potential of bacterial Xenorhabdus and Photorhabdus metabolites in the suppression of some soil-borne soybean fungal pathogens. Several secondary metabolites were identified and tested, resulting in validation of fabclavine and xenocoumacins as bioactive antifungal compounds. Overall, the study was technically rigorous, generating robust data for statistical analysis. Experimental data support the main conclusion, offering an insight into the biocontrol potential of those compounds against soybean fungal pathogens. Although well organized, the manuscript needs a revision for improved clarity, conciseness and readability.

Suggestions:

  1. The Introduction section can be expanded to give more information on the limitations of current biological control agents, particularly addressing the pain points in controlling soybean fungal diseases and a need for the innovation sought by the present study.
  2. It is also needed to expand a discussion on possible interaction mechanisms between the bioactive compounds and target pathogens (e.g., cell membrane permeability, inhibition of key enzymes) in combination with pertinent references.
  3. Table 1 is important but requires an input of SD or SE for each percent value. Please mark statistical significance with lowercase or uppercase letters alone rather than with both of them.
  4. Pay attention to F and P in italics. Two df values can be shown as subscripts of F (F<sub>df1,df2</sub>) . Keep one or two decimals (except P) throughout the manuscript.

1.The Introduction section can be expanded to give more information on the limitations of current biological control agents, particularly addressing the pain points in controlling soybean fungal diseases and a need for the innovation sought by the present study.

2.It is also needed to expand a discussion on possible interaction mechanisms between the bioactive compounds and target pathogens (e.g., cell membrane permeability, inhibition of key enzymes) in combination with pertinent references.

3.Table 1 is important but requires an input of SD or SE for each percent value. Please mark statistical significance with lowercase or uppercase letters alone rather than with both of them.

4.Pay attention to F and P in italics. Two df values can be shown as subscripts of F (F<sub>df1,df2</sub>) . Keep one or two decimals (except P) throughout the manuscript.

Author Response

Suggestions:

  1. The Introduction section can be expanded to give more information on the limitations of current biological control agents, particularly addressing the pain points in controlling soybean fungal diseases and a need for the innovation sought by the present study.

Response: A paragraph with pertinent references has been added to the introduction (L61-68) regarding the biolocigal control agents and the need for new agents.

Added references include Villavicencio-Vásquez, M., Espinoza-Lozano, F., Espinoza-Lozano, L., & Coronel-León, J. (2025). Biological control agents: mechanisms of action, selection, formulation and challenges in agriculture. Frontiers in Agronomy7, 1578915.

Galeano, R.M.S., Ribeiro, J.V.S., Silva, S.M. et al. New Strains of Trichoderma with Potential for Biocontrol and Plant Growth Promotion Improve Early Soybean Growth and Development. J Plant Growth Regul 43, 4099–4119 (2024). https://doi.org/10.1007/s00344-024-11374-z

Yu, S. F., Wang, C. L., Hu, Y. F., Wen, Y. C., & Sun, Z. B. (2022). Biocontrol of three severe diseases in soybean. Agriculture12(9), 1391.

de Faria, A. F., Schulman, P., Meyer, M. C., Campos, H. D., Cruz-Magalhães, V., Godoy, C. V., ... & Medeiros, F. H. (2022). Seven years of white mold biocontrol product’s performance efficacy on Sclerotinia sclerotiorum carpogenic germination in Brazil: a meta-analysis. Biological Control, 176, 105080.

  1. It is also needed to expand a discussion on possible interaction mechanisms between the bioactive compounds and target pathogens (e.g., cell membrane permeability, inhibition of key enzymes) in combination with pertinent references.

Response: We thank the reviewer for their comments. The discussion has been revised accordingly.

  1. Table 1 is important but requires an input of SD or SE for each percent value. Please mark statistical significance with lowercase or uppercase letters alone rather than with both of them.

Response: SE values were added. We think it is easier to show significant difference with lowercase and uppercase letters as they are.

  1. Pay attention to F and P in italics. Two df values can be shown as subscripts of F (F<sub>df1,df2</sub>) . Keep one or two decimals (except P) throughout the manuscript.

Response: All F and P were changed to italics. But we prefer giving df values separately.

Reviewer 3 Report

The article "Xenorhabdus and Photorhabdus Metabolites for Fungal Biocontrol and Application in Soybean Seed Protection", written by Nathalie Otoya-Martinez and co-authors, describes the use of new bacteria to control soybean diseases in seed treatment. The article as a whole is written and framed correctly, but there are some comments, questions and amendments to it.

  1. The article shows well the inhibitory effect of fabclavine, xenocoumacin and other compounds, but the authors limit themselves to general references to the destruction of cellular structures. It would be useful to include more data (or at least a discussion) on molecular targets and mechanisms of action. For example, at least the literature on the effect of these metabolites on cell wall synthesis or the metabolism of pathogenic fungi.
  2. Unfortunately, all the results are based on in vitro laboratory experiments and on seeds. For practical agronomy, it is important at least in the discussion to note the prospects and limitations in the transition to field conditions (the influence of soil, microbiome, competing microorganisms). Now this part is reduced to a short phrase in the conclusions, but without details.
  3. The work shows strong results against Sclerotinia sclerotiorum, Botrytis cinerea and Macrophomina phaseolina, but the block devoted to Fusarium oxysporum is much weaker (low level of inhibition, superficial discussion). We should consider in more detail (it is possible in the discussion) why Fusarium is more stable (for example, the structure of the cell wall, the release of detoxification enzymes) and how this can be overcome.
  4. The authors use the ANOVA and Tukey test, but the factors and the number of repetitions are not always clearly indicated. For example, there are no confidence intervals or standard deviations in the shelf-life section (table 1). This reduces the reproducibility and credibility of the results. It is recommended to define the design of experiments and statistical analysis more clearly.
  5. The authors mention the high stability of the metabolites during storage and compare the effectiveness with a fungicide. However, there is no discussion of the technological aspects of implementation (for example, the cost of production of bacterial metabolites, methods of large-scale production, possible regulatory restrictions, the use of chemistry to create a formulation). This is necessary to assess the real potential of the method as a biofungicide.
  6. In Figure 8, methyl thiophanate is written incorrectly, please correct the text as well.
  7. Since the metabolites of these microorganisms have a fungicidal effect, the EC50 index should be calculated and compared with fungicides. After that, the comparison with synthetic fungicides would make more logical sense.

Author Response

  1. Are the resultspresented clearly and in sufficient detail, are the conclusions supported by the results and are they put into context within the existing literature?

The article shows well the inhibitory effect of fabclavine, xenocoumacin and other compounds, but the authors limit themselves to general references to the destruction of cellular structures. It would be useful to include more data (or at least a discussion) on molecular targets and mechanisms of action. For example, at least the literature on the effect of these metabolites on cell wall synthesis or the metabolism of pathogenic fungi. The work shows strong results against Sclerotinia sclerotiorum, Botrytis cinerea and Macrophomina phaseolina, but the block devoted to Fusarium oxysporum is much weaker (low level of inhibition, superficial discussion). We should consider in more detail (it is possible in the discussion) why Fusarium is more stable (for example, the structure of the cell wall, the release of detoxification enzymes) and how this can be overcome.

Response: The discussion section has been revised as recommended

The authors mention the high stability of the metabolites during storage and compare the effectiveness with a fungicide. However, there is no discussion of the technological aspects of implementation (for example, the cost of production of bacterial metabolites, methods of large-scale production, possible regulatory restrictions, the use of chemistry to create a formulation). This is necessary to assess the real potential of the method as a biofungicide.

Response: A new paragraph has been added to discussion section as referee suggested.

Are all figures and tables clear and well-presented?

In Figure 8, methyl thiophanate is written incorrectly, please correct the text as well.

Response: Corrected.

Major comments

The article "Xenorhabdus and Photorhabdus Metabolites for Fungal Biocontrol and Application in Soybean Seed Protection", written by Nathalie Otoya-Martinez and co-authors, describes the use of new bacteria to control soybean diseases in seed treatment. The article as a whole is written and framed correctly, but there are some comments, questions and amendments to it.

Detailed comments

  1. The article shows well the inhibitory effect of fabclavine, xenocoumacin and other compounds, but the authors limit themselves to general references to the destruction of cellular structures. It would be useful to include more data (or at least a discussion) on molecular targets and mechanisms of action. For example, at least the literature on the effect of these metabolites on cell wall synthesis or the metabolism of pathogenic fungi.

Response: The precise mode of action has not been eludicated. Current theories suggest that the MOA of these compounds are through membrane leakage and RNA inhibition. This was based on current literature.

  1. Unfortunately, all the results are based on in vitro laboratory experiments and on seeds. For practical agronomy, it is important at least in the discussion to note the prospects and limitations in the transition to field conditions (the influence of soil, microbiome, competing microorganisms). Now this part is reduced to a short phrase in the conclusions, but without details.

Response: A new paragraph was added to the end of discussion.

  1. The work shows strong results against Sclerotinia sclerotiorum, Botrytis cinerea and Macrophomina phaseolina, but the block devoted to Fusarium oxysporum is much weaker (low level of inhibition, superficial discussion). We should consider in more detail (it is possible in the discussion) why Fusarium is more stable (for example, the structure of the cell wall, the release of detoxification enzymes) and how this can be overcome.

Response: This section has been improved and new references have been added Naqvi, S. A. H., Farhan, M., Ahmad, M., Kiran, R., Shahbaz, M., Abbas, A., ... & Sathiya Seelan, J. S. (2025). Fungicide resistance in Fusarium species: exploring environmental impacts and sustainable management strategies. Archives of Microbiology, 207(2), 31.

 Menna, A., Dora, S., Sancho-Andrés, G., Kashyap, A., Meena, M. K., Sklodowski, K., ... & Sánchez-Rodríguez, C. (2021). A primary cell wall cellulose-dependent defense mechanism against vascular pathogens revealed by time-resolved dual transcriptomics. BMC biology19(1), 161.

  1. The authors use the ANOVA and Tukey test, but the factors and the number of repetitions are not always clearly indicated.

Response: Data analyses section has been revised as: The inhibition percentage data were transformed using the arcsin √(x/100) function, where x represents the percentage of each treatment’s replicate. The data on mycelial growth area were analyzed using ANOVA. The main factors taken into consideration were the different bacterial strains (5 total), the four fungal pathogens, and the three supernatant concentrations (5%, 10%, and 20%). Treatment means were compared using Tukey's test at a 5% significance level. Statistical analyses were performed using SPSS version 23 and SISVAR DEX/UFLA, version 5.6.

  1. For example, there are no confidence intervals or standard deviations in the shelf-life section (table 1). This reduces the reproducibility and credibility of the results. It is recommended to define the design of experiments and statistical analysis more clearly.

Response:  In the tables Mean±SD has been added as suggested

  1. The authors mention the high stability of the metabolites during storage and compare the effectiveness with a fungicide. However, there is no discussion of the technological aspects of implementation (for example, the cost of production of bacterial metabolites, methods of large-scale production, possible regulatory restrictions, the use of chemistry to create a formulation). This is necessary to assess the real potential of the method as a biofungicide.

Response:   A paragraph has been included to discussion text:While the potential is clear, three key hurdles must be addressed to transform these metabolites into a commercially viable biofungicide. The high cost and complexity of producing and purifying bacterial metabolites must be overcome. For the product to be cost-competitive with chemical alternatives, large-scale production must be scalable and economically feasible. A stable formulation is essential. The raw metabolites must be combined with adjuvants and other inert ingredients to ensure the final product has a long shelf-life and is stable for effective field application. The most significant barrier is the rigorous, multi-year, and multi-million-dollar process of gaining regulatory ap-proval. This requires extensive toxicological and environmental studies to prove the product's safety for people and the ecosystem.”

  1. In Figure 8, methyl thiophanate is written incorrectly, please correct the text as well.

Response: Corrected.

  1. Since the metabolites of these microorganisms have a fungicidal effect, the EC50 index should be calculated and compared with fungicides. After that, the comparison with synthetic fungicides would make more logical sense.

Response: We thank the reviewer for the insightful comment. While we agree that an EC50 comparison would be valuable, the study's design used a single dose of the synthetic fungicide based on the manufacturer's recommended application rate. Therefore, we were unable to calculate a complete dose-response curve or the EC50 value for the positive control. We believe our comparison of the bacterial metabolites to the commercially relevant dose of the fungicide still provides a valid and practical assessment of their potential as biocontrol agents.

Round 2

Reviewer 3 Report

The authors have corrected most of my comments and the quality of the article has improved markedly. Thank you"

-